# Application of a Novel Biosensor for Salivary Conductivity in Detecting Chronic Kidney Disease

**DOI:** 10.3390/bios12030178

**Published:** 2022-03-17

**Authors:** Chen-Wei Lin, Yuan-Hsiung Tsai, Yen-Pei Lu, Jen-Tsung Yang, Mei-Yen Chen, Tung-Jung Huang, Rui-Cian Weng, Chun-Wu Tung

**Affiliations:** 1School of Medicine, College of Medicine, Chang Gung University, Taoyuan 33302, Taiwan; toddgod7@cgmh.org.tw; 2Department of Medical Education, Chang Gung Memorial Hospital, Chiayi 61363, Taiwan; 3Department of Diagnostic Radiology, Chang Gung Memorial Hospital, Chiayi 61363, Taiwan; russell@cgmh.org.tw; 4College of Medicine, Chang Gung University, Taoyuan 33302, Taiwan; yljwty@cgmh.org.tw (J.-T.Y.); donaldhuang@cgmh.org.tw (T.-J.H.); 5Taiwan Instrument Research Institute, National Applied Research Laboratories, Hsinchu 30261, Taiwan; ypl@narlabs.org.tw (Y.-P.L.); cian@tiri.narl.org.tw (R.-C.W.); 6Department of Neurosurgery, Chang Gung Memorial Hospital, Chiayi 61363, Taiwan; 7Department of Nursing, Chang Gung University of Science and Technology, Chiayi 61363, Taiwan; meiyen@mail.cgust.edu.tw; 8Department of Internal Medicine, Chang Gung Memorial Hospital, Yunlin 63862, Taiwan; 9Department of Respiratory Care, Chang Gung University of Science and Technology, Chiayi 61363, Taiwan; 10Graduate Institute of Biomedical Electronics and Bioinformatics, National Taiwan University, Taipei 106319, Taiwan; 11Department of Nephrology, Chang Gung Memorial Hospital, Chiayi 61363, Taiwan

**Keywords:** chronic kidney disease, salivary conductivity, non-invasive, portable, biosensor

## Abstract

The prevalence of chronic kidney disease (CKD) is increasing, and it brings an enormous healthcare burden. The traditional measurement of kidney function needs invasive blood tests, which hinders the early detection and causes low awareness of CKD. We recently designed a device with miniaturized coplanar biosensing probes for measuring salivary conductivity at an extremely low volume (50 μL). Our preliminary data discovered that the salivary conductivity was significantly higher in the CKD patients. This cross-sectional study aims to validate the relationship between salivary conductivity and kidney function, represented by the estimated glomerular filtration rate (eGFR). We enrolled 214 adult participants with a mean age of 63.96 ± 13.53 years, of whom 33.2% were male. The prevalence rate of CKD, defined as eGFR < 60 mL/min/1.73 m^2^, is 11.2% in our study. By multivariate linear regression analyses, we found that salivary conductivity was positively related to age and fasting glucose but negatively associated with eGFR. We further divided subjects into low, medium, and high groups according to the tertials of salivary conductivity levels. There was a significant trend for an increment of CKD patients from low to high salivary conductivity groups (4.2% vs. 12.5% vs. 16.9%, *p* for trend: 0.016). The receiver operating characteristic (ROC) curves disclosed an excellent performance by using salivary conductivity combined with age, gender, and body weight to diagnose CKD (AUC equal to 0.8). The adjusted odds ratio of CKD is 2.66 (95% CI, 1.10–6.46) in subjects with high salivary conductivity levels. Overall, salivary conductivity can serve as a good surrogate marker of kidney function; this real-time, non-invasive, and easy-to-use portable biosensing device may be a reliable tool for screening CKD.

## 1. Introduction

Chronic kidney disease (CKD) is one of the most important public health issues and it brings an enormous socioeconomic burden [1]. The prevalence of CKD is increasing, and it affects 8–16% of the general population worldwide currently [2,3,4]. The progression of CKD is not only associated with end-stage renal disease but also is responsible for a wide range of morbidity and mortality. Decreased renal function is also related to longer hospitalization and poor quality of life [5,6,7]. CKD is estimated to become the fifth leading cause of death globally by 2040 [8]. It is well known that early identification of CKD with timely intervention plays a crucial role in preventing disease progression [9]. However, several studies have found that greater than 90% of patients with stage 3 CKD were not aware of their disorder [3,10]. The current diagnosis or staging of CKD is based on the estimated glomerular filtration rate (eGFR) which is calculated from serum creatinine [11,12]. The need for invasive procedures to collect serum creatinine is a great limitation for screening and monitoring of CKD outside the hospital. Since most patients with early-stage CKD is asymptomatic, it makes the early diagnosis difficult if a blood test is not performed [13]. In addition, blood collection may cause short-term complications such as hematomas after venipuncture, vasovagal fainting, or phlebitis [14]. Therefore, developing an easy-to-use, non-invasive, and cost-effective biodevice for the detection of CKD is essential for large-scale screening and reducing undiagnosed CKD.

Saliva is a kind of body fluid that can be obtained easily in a non-invasive manner and serves as a potential specimen for monitoring our health situation [15,16,17,18]. The saliva has complex compositions, such as electrolytes, cytokines, enzymes, antibodies, metabolites, etc. [19]. The complexity of salivary components renders it a promising diagnostic fluid to reflect the biological changes of systemic diseases. Recent literature has demonstrated the usefulness of saliva as a monitoring biological indicator in several diseases [20,21,22]. Moreover, the area of using salivary biomarkers for screening CKD is attracting considerable interest because it can be performed non-invasively, repeatedly without trained personnel in a low-resource setting. An increasing number of clinical studies have tried to adopt saliva as a diagnostic material for CKD [14,23,24,25,26,27]. Most of these studies focused on the usage of salivary urea nitrogen or salivary creatinine levels to detect CKD. In early-stage CKD, reduced permeability of salivary gland cells and reduced plasma-to-saliva gradient would lead to incorrect measurement of salivary creatinine level. Compared to salivary creatinine, salivary urea has been shown as a more sensitive marker for CKD, particularly in earlier stages [23]. Although some studies reported that salivary urea was a useful screen tool for CKD [26,27], a cohort study by Evans et al. showed suboptimal diagnostic performance of salivary urea with the area under the receiver operating characteristic (ROC) curve equal to 0.61 [23]. Therefore, our research team has been committed to exploiting a portable, miniaturized, and ready-to-use saliva detection instrument, and trying to explore more reliable diagnostic biomarkers for CKD.

The composition of saliva changes with progressive CKD [23]. Salivary concentrations of urea nitrogen, creatinine, and most electrolytes, except calcium, are significantly higher in CKD patients than in healthy controls. The change of pH, increase in electrolytes, and accumulation of uremic particles may contribute to the alteration of the electrical conductivity of saliva. We recently fabricated a novel biodevice with miniaturized sensing probes for measuring salivary conductivity [28]. The features of this portable sensing system include a disposable printed-circuit-board (PCB) electrode and the usage of highly biocompatible, stable, and reusable gold as the conductive material. Additionally, with the co-planar design of coating-free gold electrodes, the conductivity test could be achieved with a saliva specimen at an extremely low volume (50 μL). This reduces the difficulty of collecting saliva, eliminates the need for trained personnel, and increases people’s willingness to the test. Our previous study has shown that the increase in salivary conductivity was associated with the increase in serum and urinary osmolality in dehydrated healthy adults [29]. Moreover, we also noticed that age and serum osmolality correlated well with salivary conductivity in hemodialysis patients [30]. Our preliminary data including 10 CKD patients and 10 healthy controls discovered that the salivary conductivity was significantly higher in the CKD population [28]. Although this result is encouraging, it still needs to be confirmed by large-scale studies. Consequently, this pilot study aims to validate the correlation between salivary conductivity and eGFR in a healthy population and to investigate whether the biodevice can act as a useful tool for screening and detecting CKD.

## 2. Materials and Methods

### 2.1. The Sensing Device and System

The portable biodevice was implemented to measure salivary conductivity. In brief, the construction steps were composed of three main steps, including biodevice design, the collection and analysis of saliva, and the test of reusability and selectivity of the electrodes.

#### 2.1.1. Design of the Biodevice for Measuring Conductivity

The proposed system is composed of two main parts, including a PCB with coating-free gold electrodes and a conductivity meter (Figure 1). The size of the micro-fabricated electrode is 2 × 2 mm^2^ which can be covered with testing samples with only 50 μL required. The 10 × 5.5 × 2.2 cm^3^ conductivity meter was fabricated as the previous study, which composes of an analog-to-digital converter (ADC) (AD5933, Analog Devices, Norwood, MA, USA), a micro control unit (MCU system) (STMicroelectronics, Geneva, Switzerland), a temperature sensor (Aosong, Guangzhou, China), and an organic light-emitting diode (OLED) (Zhongjingyuan, Henan, China). The electrode was bonded to the PCB by using nickel immersion gold wired. The conductivity meter can be implemented with 1 Vpp and 1 kHz sine waves via ADC to accomplish the measurement. The electrical conductivity parameters can be acquired through a discrete Fourier transform. The conductivity signal at 25 °C can be calculated through temperature compensation by the MCU system and be displayed on the OLED within 10 s.

#### 2.1.2. The Collection and Analysis of Saliva

Salivary samples of all eligible subjects were collected and processed as previously described [28,29]. Briefly, the participants were asked to swallow several times for emptying their mouths before collection. The saliva was then collected by placing a mouth care cotton swab (diameter = 0.9 cm, length = 15.24 cm) under the tongue for 2 min. The collected salivary specimens were loaded into the well of the sensing probe and then analyzed for electrical conductivity through the conductivity meter which was connected to the portable device through a USB port (Figure 2). With the design of coplane, miniaturized, and coating-free gold electrodes, the conductivity test could be achieved with a saliva specimen at an extremely low volume (50 μL). A minimum measurement volume of 50 uL was determined according to the area of the electrodes, as this is the appropriate sample volume to fully cover the surface of micro-electrodes. Furthermore, stable conductivity data can be obtained when evaluating under this condition. The conductivity meter was pre-calibrated using the standard conductivity solution and examined coefficient of variation was less than 1%. The detection time including sample preparation is within 5 min. The saliva solution is quite stable during the period. We had demonstrated that there was no significant change in the conductivity measurement of salivary samples at room temperature for at least 15 min (Appendix A).

#### 2.1.3. Reusability of the Electrode

For testing the reusability of the electrode, salivary samples were tested 20 times with the same electrode. We used ultrapure water to clean the well until confirming the conductivity drops to the same level as the baseline before every measurement. In the clinical study, each PCB electrode was used with no more than five salivary samples to ensure the quality of the clinical data.

#### 2.1.4. Selectivity of the Sensor

Saliva is composed of 99% water, 1% protein and salts, etc. [31]. Our sensor is used to measure the salivary conductivity which is determined by the electrical admittance between the electrodes which mostly reflects the concentration of the electrolytes. To increase the selectivity of the sensor and eliminate the interfering factors, we have adopted some special designs for our device. First, since the signal obeys the path of least impedance, the electro-path of electro-impedance spectroscopy is mainly at the edge of microelectrodes, which is approved according to the previous study [32]. Therefore, most of the interference effect caused by large particles such as food debris and nasal secretion, which fall at the top of the co-planar electrode can be minimized and have better selectivity. Second, the protein in saliva may decrease electrode sensitivity for detecting electrolytes because some proteins will adhere to the electrodes. However, with our microfabricated co-planar electrodes design, the interfering factor can be decreased compared to the commercial electrodes. This can be explained by the fact that the microelectrode edge is less vulnerable to protein adhesions than the electrode surface. To justify the hypothesis of our design, we have conducted experiments that measured the conductivity of ultrapure water, bovine serum albumin (BSA), phosphate-buffered saline (PBS) solution, different BSA concentrations mixed with PBS solution, and healthy participants’ saliva. According to Lin et al., BSA can be a surrogate of the proteins secreted by the salivary glands, which is optimal for testing the selectivity of our sensor, and the mean value of human salivary protein ranged from 0.72 to 2.45 mg/mL [33]. In addition, PBS solution can be a surrogate of the electrolytes. The conductivity of each solution was measured six times repeatedly and the average was used to compare the difference between the groups. Besides, other experiments were conducted to compare our microelectrode with the commercial electrode by detection of ultrapure water spiked in with BSA to determine whether our microfabricated co-planar electrodes design has less interference with protein factors. Third, because interference particles can accumulate on the surface of the salivary solution with time, we further experimented to test the stability of the saliva sample. The conductance of six different salivary samples was repeatedly tested for 15 min with an interval of 3 min. The serial change can also reflect the stability of the saliva. Fourth, only a 50 μL saliva sample is required to perform the test. Therefore, we can assume that the sample temperature achieves equilibrium with the ambient temperature within a short period of time. It means that temperature is also not an interfering factor.

### 2.2. Clinical Study Design and Participants

This is a cross-sectional pilot study including adults aged ≥18 years who attended the annual health examination at Yunlin Branch of the Chang Gung Memorial Hospital, a regional teaching hospital in southern Taiwan, in August 2021. Before the investigation, we conducted a sample size calculation using PASS V.15 (NCSS, Kaysville, UT, USA). Assuming a CKD prevalence rate of 12% [2,3], with 80% power, at a two-sided statistical significance level of 5% (α = 0.05), the sample size needed for an acceptable area under the ROC curve of 0.70 is 171. During the study period, 241 consecutive subjects completed the general health examinations. Among them, nine subjects could not cooperate with the saliva collection. Another 18 subjects were excluded owing to active illness, recent hospitalization for acute diseases, or past histories of head and neck treatments, such as surgery or radiotherapy. Therefore, a total of 214 adult subjects were included in the final analysis, exceeding the sample size required for the desired study power. The post hoc power analysis demonstrated a sample size of 214 achieved 99% power for salivary conductivity plus age and 80% power for only salivary conductivity as diagnostic tools at a 5% significance level. This study complied with the guidelines of the Declaration of Helsinki and was approved by the Medical Ethics Committee of Chang Gung Memorial Hospital (institutional review board number: 202000109B0 and 202002186B0). Before the beginning of the study, all participants agreed and signed the informed consent form.

### 2.3. Procedures of Clinical Study

Before health examination, all subjects received a comprehensive questionnaire survey conducted by trained nurses to provide information about previous head and neck surgery or radiotherapy and comorbid diseases, including diabetes, hypertension, chronic kidney disease, ischemic heart disease, stroke, dyslipidemia, gout, and chronic liver disease. Anthropometric data of their body weight (kg) and height (cm) were obtained, and the corresponding body mass index (BMI) was calculated as weight in kilograms divided by height in meters squared (kg/m^2^). The blood pressure was measured with a validated electronic automated sphygmomanometer on the right arm of seated participants after 15 min’ rest in a quiet and comfortable environment. Two blood pressure readings were obtained at an interval of 5 min; if the readings differed by more than 20 mmHg, a third measurement was made, and the two closest blood pressure values were averaged. After an 8 to 12 h overnight fasting, venous blood and salivary samples were collected. Blood specimens were left undisturbed at room temperature for 30 min to allow the blood to clot. The clot was then removed by centrifuging at 3000 rpm for 10 min. The resulting supernatant was the serum which was analyzed for biochemical data using an automatic chemistry analyzer (Beckman DXC880i, Brea, CA, USA) following standardized laboratory procedures.

### 2.4. Definition of Chronic Kidney Disease

We used the revised Modification of Diet in Renal Disease (IDMS-MDRD) equation to calculate the eGFR: 175 × Cr^−1.154^ × Age^−0.203^ × (0.742, if female) [34]. The definition of chronic kidney disease is the presence of an eGFR less than 60 mL/min/1.73 m^2^.

### 2.5. Statistical Analysis

Continuous variables are expressed as means ± standard deviations and categorical variables are displayed as numbers with their percentages. For examining the normality of numerical variables, the Kolmogorov–Smirnov method was performed. The independent two-tailed Student’s *t*-test was used to compare the means of the continuous variables of normal distribution, while the Mann–Whitney *U* test was applied for continuous but not normally distributed data. For comparison between multiple groups, the Analysis of Variance (ANOVA) with post hoc analysis were used for quantitative variables. The Pearson’s chi-square test was conducted for the comparison of categorical variables. The association between salivary conductivity and other quantitative variables was analyzed by univariate linear regression models. Furthermore, the multivariate linear regression analysis with backward selection was utilized to form a predictive model for salivary conductivity. Statistically significant factors identified in the univariate analysis were incorporated into the multivariate regression analysis. ROC curve analysis was used to evaluate the diagnostic accuracy of salivary conductivity on CKD. To improve the diagnostic ability, we performed the multivariable logistic regression analysis with backward selection using combined variables. The diagnostic power of different models was determined by calculating the area under the ROC curve (AUROC). Multivariate logistic regression with stepwise backward selection was utilized to estimate the odds ratios (ORs) of CKD, with salivary conductivity and other risk factors as independent variables. Subgroup analyses for heterogeneity in saliva conductivity effect were further performed, with subgroups defined according to age (<75 years or ≥75 years), gender, BMI (<25 kg/m^2^ or ≥25 kg/m^2^), history of diabetes (yes or no), underlying hypertension (yes or no), and coexisting dyslipidemia (yes or no). All statistical analyses were two-sided and were performed using the Statistical Program for Social Sciences (SPSS) version 22 (IBM Corporation, Armonk, NY, USA). Values of *p* < 0.05 were considered statistically significant.

## 3. Results

### 3.1. Reusability of the Sensor

After serial measurements of the salivary conductivity of the same saliva sample 20 times, the mean absolute percentage error (MAPE) is 0.88%, with a maximum error of 2.39% (Figure 3).

### 3.2. Selectivity of the Sensor

For determining the selectivity of the sensor, we measured the conductivity of ultrapure water, BSA, PBS solution, different BSA concentrations mixed with PBS solution, and healthy participants’ saliva. The results revealed that our co-planar microelectrodes have almost no interference from BSA and the majority of the conductivity was originated from the electrolytes in the solution (Figure 4A). By detection of ultrapure water spiked in with BSA, we noticed that the slope of the commercial electrode is three times significantly higher than our tailor-made design, which means our design is less vulnerable to BSA influence (Figure 4B). In addition, the conductivity of the saliva samples was stable for at least 15 min, which indicates that the proposed device has great resistance to interference particles accumulation (Figure 4C).

### 3.3. Demographic Characteristics of Study Participants

This cross-sectional study included 214 subjects with a mean age of 63.96 ± 13.53 years, of whom 33.2% were male. The mean salivary conductivity value was 5.91 ± 1.79 ms/cm. Diabetes mellitus, hypertension, chronic kidney disease, ischemic heart disease/stroke, dyslipidemia, gout, and chronic liver disease were recognized in 26 (12.1%), 62 (29.0%), 3 (1.4%), 16 (7.5%), 30 (14.0%), 9 (4.2%) and 24 (11.2%) participants, respectively. Their mean eGFR was equal to 86.33 ± 22.81 mL/min/1.73 m^2^. Other biochemical parameters, anthropometric data, and blood pressure were shown in Table 1.

### 3.4. Association between Salivary Conductivity and Clinical Variables

The relationship between salivary conductivity and continuous variables was analyzed by Pearson’s correlation coefficient. Salivary conductivity was positively associated with age, systolic blood pressure, blood urea nitrogen (BUN), creatinine, and fasting glucose, but was inversely related to the eGFR (Table 1).

We then conducted the backward multivariate linear regression analyses to determine the salivary conductivity level from clinical parameters. Only Significant factors identified in the simple regression analysis were applied to the multivariate analysis. A co-linearity analysis showed all tolerance > 0.1 and variance inflation factor (VIF) < 5, thus no co-linearity exists among independent variables (Appendix A). Adjusted variables in regression model 1 consisted of age, systolic blood pressure, BUN, fasting glucose, and eGFR. Only older age (standardized β = 0.264, *p* < 0.001), higher fasting glucose (standardized β = 0.132, *p* = 0.037), and lower eGFR (standardized β = −0.186, *p* = 0.010) were significantly associated with higher salivary conductivity (R^2^ = 0.176, model 1 in Table 2). Because eGFR is calculated from serum creatinine, we replaced eGFR with creatinine in regression model 2. It is noted that creatinine was still positively correlated with salivary conductivity (standardized β = 0.151, *p* = 0.024, model 2 in Table 2).

### 3.5. The Prevalence of Chronic Kidney Disease (CKD) Increases with Salivary Conductivity

To investigate the association of salivary conductivity and CKD, we divided study subjects into low, medium, and high salivary conductivity groups based on the tertial of conductivity levels (Figure 5). There was a significant trend for an increment of CKD prevalence rate from low to high salivary conductivity levels (4.2% in low vs. 12.5% in middle vs. 16.9% in high conductivity group, *p* for trend = 0.016).

### 3.6. The Use of Salivary Conductivity to Detect Individuals with CKD

To examine the diagnostic ability of salivary conductivity on CKD, the ROC curve analysis was conducted. The AUROC was equal to 0.648 (95% CI: 0.542–0.755) when salivary conductivity is the only predicting factor. To improve the diagnostic performance, age, gender, and body weight were further combined into the prediction model. This combination model demonstrated a significant increase in AUROC to 0.798 (95% CI: 0.726–0.871) (Figure 6A). Strikingly, the AUROC would be 0.751 (95% CI: 0.577–0.926) with salivary conductivity as the only predictor if the study group was restricted to individuals ≥75 years old (Figure 6B).

### 3.7. Characteristics of Low Versus High Salivary Conductivity Population

To further evaluate the risk of CKD concerning salivary conductivity, the subjects were stratified into low and high salivary conductivity level groups based on the cutoff value of the ROC curve calculated by using only salivary conductivity as the predictor (6.59 ms/cm). The characteristics of low versus high salivary conductivity groups were summarized in Table 3. The mean salivary conductivity of the low and high salivary conductivity groups was 4.84 ± 1.01 and 7.94 ± 1.02 ms/cm, respectively. It was noticed that subjects with high salivary conductivity were older (69.92 ± 10.93 vs. 60.81 ± 13.75 years, *p* < 0.01). They also had higher systolic blood pressure (135.68 ± 21.91 vs. 128.93 ± 19.78 mmHg, *p* = 0.02), higher BUN (16.97 ± 5.78 vs. 14.34 ± 4.77 mg/dL, *p* < 0.01), higher creatinine (0.90 ± 0.30 vs. 0.76 ± 0.22 mg/dL, *p* < 0.01), lower eGFR (77.17 ± 21.27 vs. 91.18 ± 22.17 mL/min/1.73 m^2^, *p* < 0.01), higher serum osmolality (288.69 ± 6.15 vs. 287.16 ± 5.04 mOsm/kgH_2_O, *p* = 0.05), higher fasting glucose level (108.16 ± 20.34 vs. 100.41 ± 27.27 mg/dL, *p* < 0.01), higher hemoglobin A1c (5.97 ± 0.83 vs. 5.85 ± 1.05%, *p* = 0.03), and a higher percentage of underlying diabetes mellitus (20.5 vs. 7.9%, *p* < 0.01). There were no significant differences in gender, body weight and height, body mass index, diastolic blood pressure, the prevalence of hypertension, chronic kidney disease, ischemic heart disease/stroke, dyslipidemia, gout, and chronic liver disease, ALT, triglyceride, total cholesterol, LDL-C, and HDL-C between groups.

### 3.8. Subgroup Analysis of the Risk of CKD, Comparing High versus Low Salivary Conductivity

The subgroup analysis of the risk of CKD, comparing high versus low salivary conductivity was shown in Figure 7. Overall, the risk of CKD was higher in the high salivary conductivity group, with an adjusted odds ratio of 2.66 (95% CI, 1.10–6.46). There was no significant difference in the risk for CKD among subgroups of age and BMI. In the male population, and those without DM, without hypertension, and without dyslipidemia, the risk of CKD was significantly higher in the high salivary conductivity group, with the odds ratio of 4.60 (95% CI, 1.01–21.04), 2.99 (95% CI, 1.15–7.73), 3.71 (95% CI, 1.11–12.37), and 3.29 (95% CI, 1.27–8.55), respectively. To avoid bias introduced by unequal distribution of confounding variables, we further conducted sensitivity analyses in subjects without histories of diabetes and hypertension (Appendix A) and used a propensity score-matched dataset. The results were in concordance with the original analysis (Appendix A).

## 4. Discussion

The results reported here reveal that salivary conductivity is negatively correlated with eGFR and can serve as a potential biomarker for detecting CKD. Traditionally, the golden standard of measuring kidney function requires the collection of 24 h urine creatinine amount and the blood test of serum creatinine, which is cumbersome for patients [11,35]. Simplified estimation of renal function by MDRD equation still needs an invasive venipuncture. As mentioned in the Results section, there was a notably negative correlation between salivary conductivity and eGFR in the simple and multivariate Pearson’s correlation analyses (Table 1 and Table 2). In addition, there was an increasing trend of CKD prevalence from low to high salivary conductivity levels (Figure 5). Hence, the salivary conductivity may be an alternative surrogate marker for kidney function.

The prevalence rate of CKD, defined as eGFR < 60 mL/min/1.73 m^2^ by the revised MDRD equation, is 11.2% in our study. A large-scale cohort study based on 462,293 adults in Taiwan showed that the national prevalence of CKD was 11.93% [3]. This indicates the representativeness of our research subjects. In addition, only three participants in our study answered that they had CKD by the questionnaire. It revealed that the awareness of CKD in our study population was very low (12.5%). In line with this observation, previous research has also pointed out the serious problem of unawareness in the management of CKD, especially in patients with early-stage CKD [3,10,36]. Since saliva collection is an easier and non-invasive method, this makes large-scale screening or continuous self-monitoring of CKD less difficult.

Furthermore, multivariate linear regression analyses also showed a significantly positive correlation between salivary conductivity with age or fasting glucose (Table 2). Subjects with high salivary conductivity were prone to be older and had higher fasting sugar, higher hemoglobin A1c, and a higher percentage of diabetic history (Table 3). Several studies have found that the secretion and properties of saliva change with age [37,38]. Similar to CKD patients, most electrolytes increase in the saliva of healthy elderly individuals [23,37], which may contribute to the increase in salivary conductivity. Recent studies have also shown a strong association between salivary conductivity and age [30,39]. Further, salivary glucose concentrations were proved to be related to serum glucose levels [40,41]. Since CKD patients are mostly the elderly and prone to develop hyperglycemia [42], this may partly explain why they have higher salivary conductivity. Besides, we also noticed a significant correlation between systolic blood pressure and salivary conductivity. Subjects in the high salivary conductivity group were shown to have higher systolic blood pressure. A recent report by Labat et al. has shown that salivary electrolytes increased with age and were associated with hypertension [39], which was compatible with our findings.

The ROC curve of our study implied that using salivary conductivity combined with age, gender, and body weight can yield an excellent prediction model with AUC equal to 0.8 [43]. The traditional concept of the elderly is defined as an age of 65 years or older. However, many of the elderly, especially those aged younger than 75 years, are still healthy and active due to the progress of modern medicine and nutritional support. Ouchi et al. had redefined aged from 65 to 74 years as pre-elderly and aged over 75 years as the new definition of elderly [44]. In addition, aging is a significant risk factor for developing CKD. Mallappallil et al. had demonstrated that the patients aged from 75 to 79 years had a 40% higher risk of having CKD than those aged from 65 to 74 years. [45]. Therefore, we further performed ROC curve and logistic regression analyses among patients greater than 75 years. Although using only salivary conductivity to predict CKD was not good enough with AUC only equal to 0.65, we found the model could perform well in the subgroup of individuals older than 75 years with AUC equal to 0.75 (Figure 6). This finding is consistent with the result in Figure 7 that older patients with higher salivary conductivity had a higher odds ratio to have CKD. A previous finding that salivary electrolytes concentration increases with age [39] provides support for these results. The morphology and function of nephrons are affected by the aging process, which causes the elderly vulnerable to acute or chronic kidney damages [46]. The prevalence of CKD is markedly higher in elder populations [47,48]. The property of better diagnostic performance in the high-risk elderly makes salivary conductivity a promising tool for CKD screening.

Figure 7 shows a strong association of salivary conductivity with the risk of CKD. Although there was a tendency for higher blood pressure and higher fasting blood glucose in subjects with high salivary conductivity (Table 1 and Table 3), the subgroup analyses demonstrated that salivary conductivity distinguished CKD better among subjects without diabetes, hypertension, and dyslipidemia (Figure 7). One possible explanation for this discrepancy was that subjects with these comorbidities were little in number, 26, 72, and 30 respectively, which created a wider confidence interval. Despite that higher salivary conductivity in the patients older than 75 years did not have a significantly higher risk of CKD, there was a trend of increasing risk for CKD compared to those younger than 75 years old. However, the phenomenon of better discrimination of CKD in healthier subjects may represent salivary conductivity as a good diagnostic tool. Further analysis with a larger population is warranted to clarify these findings. In addition, patients with CKD usually had many comorbidities such as diabetes and hypertension. Salivary conductivity may merely reflect the comorbidities but not kidney function. However, we can find that those patients with higher salivary conductivity but without diabetes or hypertension still had a higher risk of having CKD. This finding can again suggest our conclusion that salivary conductivity has a positive correlation with CKD.

There are several limitations needed to be mentioned in this research. First, this is a cross-sectional study; therefore, the true relationship between salivary conductivity and eGFR or CKD could not be confirmed by a single test. Second, CKD in our study was defined by a single eGFR measurement, which may erroneously include subjects with acute kidney injury. Third, CKD was defined as an eGFR < 60 mL/min/1.73 m^2^, thus individuals of higher eGFR but with kidney damage were not diagnosed. This would underestimate the occurrence or risk of CKD. In addition, current criteria for diagnosing CKD need eGFR and albuminuria or proteinuria data. A lack of urinalysis, proteinuria, or albuminuria data in study participants may underestimate the prevalence of CKD. However, the percentage of CKD is 11.2% in our study, which was comparable with the reported 12% national prevalence of CKD in Taiwan, reported by Wen et al. [3]. This suggests that our research subjects are still representative. Fourth, this study was performed only in the Asian population, thus the results may not be correctly applied to other ethnicities. Besides, serum electrolytes status was not measured during the health examination. Since salivary electrolyte concentrations may vary with serum, the lack of serum electrolyte data may affect the assessment of salivary conductivity. Finally, we did not analyze the components of saliva in CKD versus non-CKD subjects. Therefore, we could not fully explain the nature of why salivary conductivity increases in CKD subjects.

## 5. Conclusions

Timely screening and detection for undiagnosed CKD are decisive for delaying the disease progression and reducing its complications or mortality. The results of this study demonstrate a significant correlation between salivary conductivity and eGFR. Higher salivary conductivity is associated with an increased risk of CKD. We have also shown an acceptable diagnostic accuracy and sensitivity of salivary conductivity in distinguishing CKD, especially for the elderly. Taken together, salivary conductivity can serve as a good surrogate marker of renal function; this real-time, non-invasive, and easy-to-use portable biosensing device may be a reliable tool for screening early-stage CKD.

## Figures and Tables

**Figure 1 biosensors-12-00178-f001:**
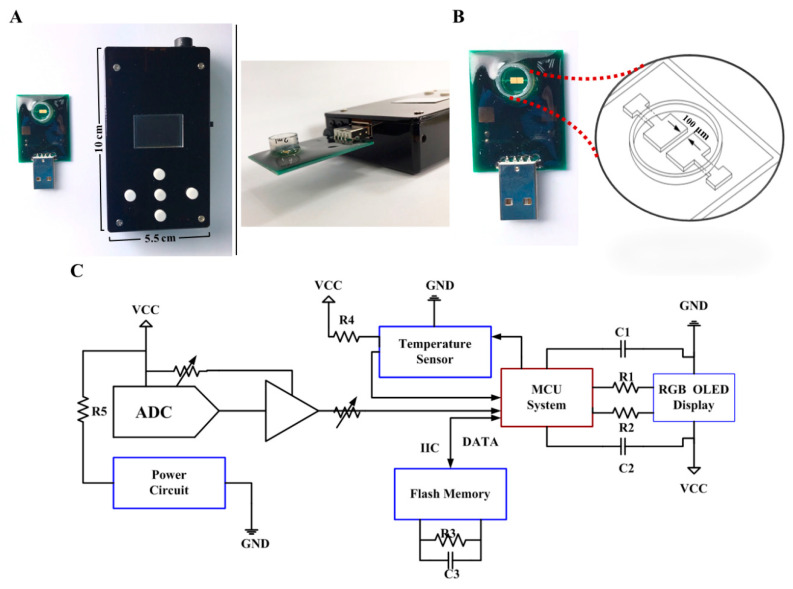
A tailor-made portable biodevice for measuring conductivity. (**A**) The device is composed of two parts, including a PCB with a coating-free gold microelectrode sensor in the sample well and a conductivity meter with a size of 10 × 5.5 × 2.2 cm^3^. (**B**) Design of the co-planar microelectrode and the sample well. The gap between two microelectrodes is only 100 μm. (**C**) Electronic schematics of the fabricated conductivity meter. Abbreviations: ADC, analog-to-digital converter; C, capacitance; GND, ground; IIC, inter-integrated circuit; MCU, micro control unit; OLED, organic light-emitting diode; R, resistance; RGB, red, green, blue color model; VCC, volt current condenser.

**Figure 2 biosensors-12-00178-f002:**
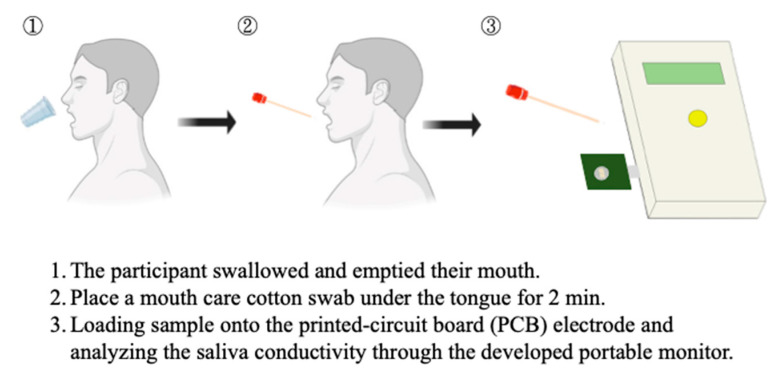
The saliva collection and salivary conductivity analysis protocol.

**Figure 3 biosensors-12-00178-f003:**
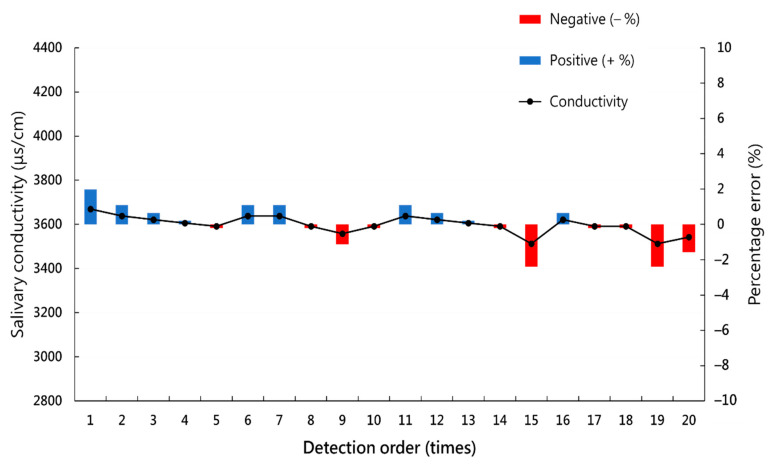
Serial measurements of the salivary conductivity of the same saliva sample 20 times.

**Figure 4 biosensors-12-00178-f004:**
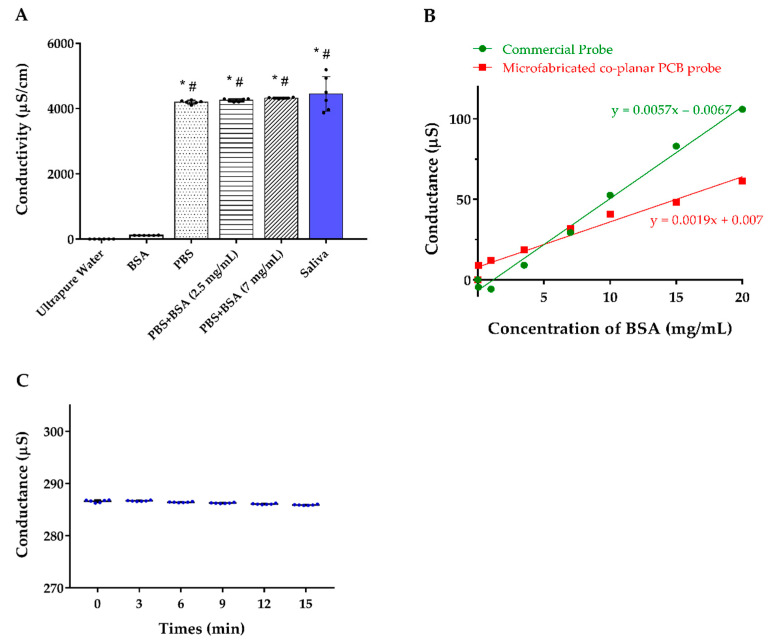
(**A**) Conductivity of the solution with different components. (**B**) Detection of the conductance of ultrapure water spiked in with BSA comparing our co-planar microelectrode and the commercial electrode. (**C**) Serial conductance measurements of the 6 individuals’ saliva at different time points. Abbreviations: BSA, bovine serum albumin; PBS, phosphate-buffered saline. * indicates *p* < 0.05 compared to ultrapure water; # indicates *p* < 0.05 compared to the BSA.

**Figure 5 biosensors-12-00178-f005:**
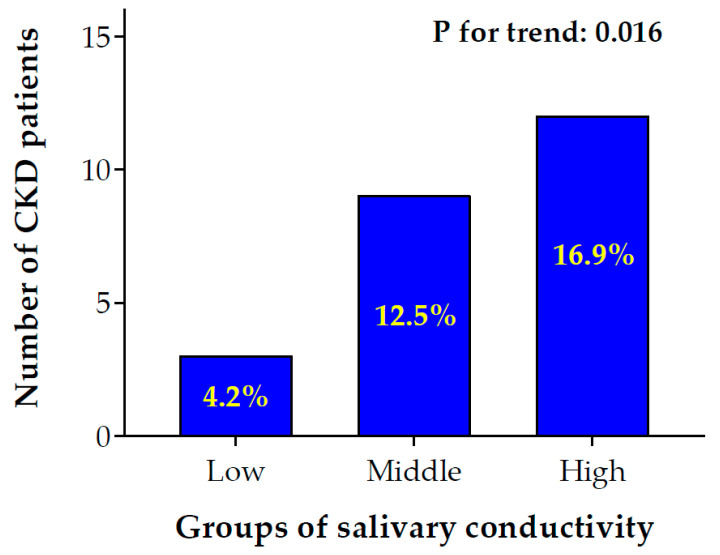
Bar chart of the number of CKD patients with low to high salivary conductivity levels. Participants were separated into low, middle, and high salivary conductivity levels based on the tertials of conductivity levels (low ≤ 4.84 ms/cm; 4.84 ms/cm < middle ≤ 6.60 ms/cm; high > 6.60 ms/cm). The number and percentage of patients with CKD were shown in the bar chart (*p* for trend: 0.016). CKD, chronic kidney disease.

**Figure 6 biosensors-12-00178-f006:**
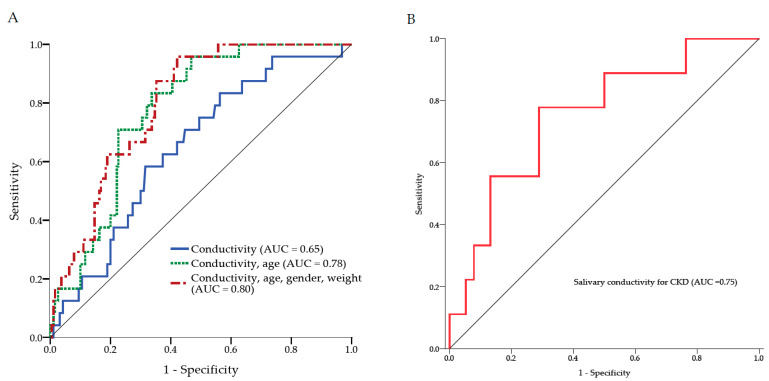
The receiver operating characteristic curve analysis. (**A**) Three ROC models for the diagnosis of CKD. The AUC was equal to 0.65 when salivary conductivity is the only predicting factor. The combination of salivary conductivity, age, gender, and body weight as the predicting factors showed the AUC was increased to 0.80. (**B**) The ROC curve analysis for the diagnostic accuracy of salivary conductivity on CKD among subjects older than 75 years. AUC, area under the ROC curve; CKD, chronic kidney disease; ROC, receiver operating characteristic.

**Figure 7 biosensors-12-00178-f007:**
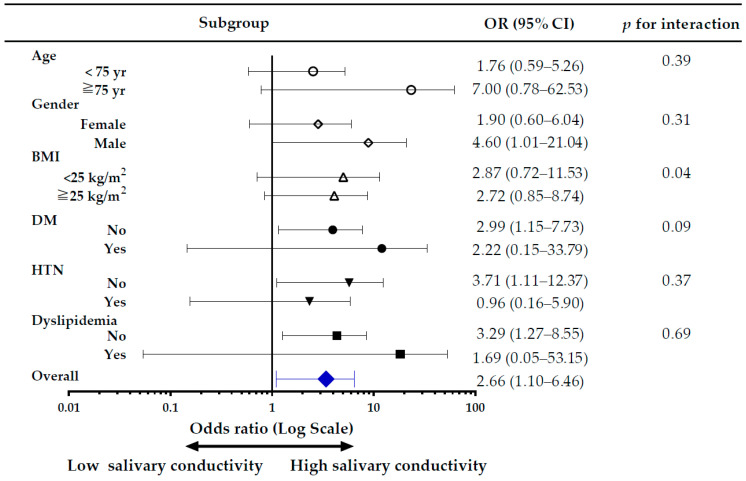
Forest plot for subgroup analysis of CKD risk. The adjusted odds ratio (OR) is for the high versus low salivary conductivity group. Study subjects were divided into high and low salivary conductivity groups based on the cutoff value of the ROC curve analysis (6.59 ms/cm). BMI, body mass index; CKD, chronic kidney disease; DM, diabetes mellitus; HTN, hypertension.

**Table 1 biosensors-12-00178-t001:** Characteristics of the study population and the correlation between salivary conductivity and continuous variables.

	All (N = 214)	Pearson *r*	*p* Value for *r*
**Salivary conductivity, ms/cm**	5.91 ± 1.79		
**Demographics**			
Age, years	63.96 ± 13.53	0.362	<0.01
Gender (male), n (%)	71 (33.2)		
Body weight, kg	63.08 ± 10.79	0.049	0.48
Body height, cm	158.97 ± 7.67	0.037	0.59
Body mass index, kg/m^2^	24.90 ± 3.36	0.027	0.69
Systolic blood pressure, mmHg	131.26 ± 20.74	0.193	<0.01
Diastolic blood pressure, mmHg	75.24 ± 11.79	0.046	0.50
**Comorbid conditions, n (%) ^@^ **			
Diabetes mellitus	26 (12.1)		
Hypertension	62 (29.0)		
Chronic kidney disease	3 (1.4)		
Ischemic heart disease/stroke	16 (7.5)		
Dyslipidemia	30 (14.0)		
Gout	9 (4.2)		
Chronic liver disease	24 (11.2)		
**Laboratory parameters**			
BUN, mg/dL	15.26 ± 5.28	0.178	<0.01
Creatinine, mg/dL	0.81 ± 0.26	0.251	<0.01
eGFR, mL/min/1.73 m^2^	86.33 ± 22.81	−0.323	<0.01
Serum osmolality, mOsm/kgH_2_O	287.69 ± 5.48	0.106	0.12
Fasting glucose, mg/dL	103.09 ± 25.31	0.153	0.03
Hemoglobin A1c, %	5.89 ± 0.98	0.045	0.51
ALT, U/L	22.10 ± 20.91	0.027	0.70
Triglyceride, mg/dL	113.19 ± 74.33	0.105	0.13
Total cholesterol, mg/dL	198.86 ± 39.48	−0.056	0.41
LDL-C, mg/dL	120.77 ± 33.63	−0.066	0.33
HDL-C, mg/dL	55.56 ± 13.22	−0111	0.11

Values are expressed as the mean ± standard deviation or number (percentage). @ The information of comorbid conditions was obtained by questionnaires. Abbreviations: ALT, alanine aminotransferase; BUN, blood urea nitrogen; eGFR, estimated glomerular filtration rate; HDL-C, high-density lipoprotein cholesterol; LDL-C, low-density lipoprotein cholesterol.

**Table 2 biosensors-12-00178-t002:** Multivariate linear regression analyses of determinants associated with the salivary conductivity.

**Model 1**	**Unstandardized Coefficients** **β (Standard Error)**	**Standardized β**	***p* Value**
Constant	3.974 (1.062)		<0.001
Age	0.035 (0.010)	0.264	<0.001
Fasting glucose	0.009 (0.004)	0.132	0.037
eGFR	−0.015 (0.006)	−0.186	0.010
**R^2^ = 0.176**			
**Model 2**	**Unstandardized Coefficients** **β (Standard Error)**	**Standardized β**	***p* Value**
Constant	1.452 (0.724)		0.046
Age	0.040 (0.009)	0.306	<0.001
Fasting glucose	0.010 (0.004)	0.141	0.026
Creatinine	1.043 (0.459)	0.151	0.024
**R^2^ = 0.170**			

Parameters included in model 1: age, systolic blood pressure, BUN, fasting glucose, and eGFR. Parameters included in model 2 are the same as in model 1, except that creatinine is used to replace eGFR. Abbreviations: BUN, blood urea nitrogen; eGFR, estimated glomerular filtration rate. The backward multivariate linear regression analysis method was conducted.

**Table 3 biosensors-12-00178-t003:** Population characteristics of low and high salivary conductivity groups.

	Low Salivary Conductivity Group * (N = 140)	High Salivary Conductivity Group (N = 74)	*p* Value
**Salivary conductivity, ms/cm**	4.84 ± 1.01	7.94 ± 1.02	<0.01 ^#^
**Demographics**			
Age, years	60.81 ± 13.75	69.92 ± 10.93	<0.01 ^#^
Gender (male), n (%)	42 (30.0)	29 (39.2)	0.17
Body weight, kg	62.94 ± 10.60	63.35 ± 11.21	0.94
Body height, cm	159.08 ± 7.39	158.76 ± 8.24	0.45
Body mass index, kg/m^2^	24.82 ± 3.34	25.07 ± 3.40	0.61
Systolic blood pressure, mmHg	128.93 ± 19.78	135.68 ± 21.91	0.02 ^#^
Diastolic blood pressure, mmHg	74.76 ± 11.60	76.14 ± 12.16	0.42
**Comorbid conditions, n (%) ^@^**			
Diabetes mellitus	11 (7.9)	15 (20.5)	<0.01 ^#^
Hypertension	36 (25.7)	26 (35.6)	0.13
Chronic kidney disease	1 (0.7)	2 (2.7)	0.27
Ischemic heart disease/Stroke	7 (5.0)	9 (12.3)	0.05
Dyslipidemia	20 (14.3)	10 (13.7)	0.91
Gout	4 (2.9)	5 (6.8)	0.17
Chronic liver disease	16 (11.4)	8 (11.0)	0.92
**Laboratory parameters**			
BUN, mg/dL	14.34 ± 4.77	16.97 ± 5.78	<0.01 ^#^
Creatinine, mg/dL	0.76 ± 0.22	0.90 ± 0.30	<0.01 ^#^
eGFR, mL/min/1.73 m^2^	91.18 ± 22.17	77.17 ± 21.27	<0.01 ^#^
Serum osmolality, mOsm/kgH_2_O	287.16 ± 5.04	288.69 ± 6.15	0.05 ^#^
Fasting glucose, mg/dL	100.41 ± 27.27	108.16 ± 20.34	<0.01 ^#^
Hemoglobin A1c, %	5.85 ± 1.05	5.97 ± 0.83	0.03 ^#^
ALT, U/L	22.89 ± 24.87	20.62 ± 9.74	0.69
Triglyceride, mg/dL	107.50 ± 62.19	123.95 ± 92.62	0.10
Total cholesterol, mg/dL	198.18 ± 36.15	200.15 ± 45.37	0.73
LDL-C, mg/dL	120.09 ± 30.79	122.05 ± 38.65	0.69
HDL-C, mg/dL	56.46 ± 12.81	53.84 ± 13.89	0.17

Values are expressed as mean ± standard deviation or number (percentage). * Study populations were stratified into low and high groups according to the cutoff value of salivary conductivity (6.59 ms/m). @ The information of comorbid conditions was obtained by questionnaires. # indicates *p*-value < 0.05. Abbreviations: ALT, alanine aminotransferase; BUN, blood urea nitrogen; eGFR, estimated glomerular filtration rate; HDL-C, high-density lipoprotein cholesterol; LDL-C, low-density lipoprotein cholesterol.

## Data Availability

The data presented in this study are available on request from the corresponding author. The data are not publicly available due to privacy.

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
