# Peer review of "Application of a Novel Biosensor for Salivary Conductivity in Detecting Chronic Kidney Disease"

_biosensors, 2022, doi:10.3390/bios12030178_

Round 1

Reviewer 1 Report

The authors propose and demonstrate a device with miniaturised coplanar biosensing probes for measuring salivary conductivity in patients with chronic kidney disease (CKD) at an extremely low volume (50 uL). I believe that revision is necessary to ensure that the Journal maintains a high standard of publication.

  1. In Section 2, authors should discuss the materials that were used to complete the proposed work. What kind of chemicals, reagents, and so forth are used in this process?
  2. What is the rationale for taking two blood pressure readings at a 5-minute interval?
  3. Why were venous blood and salivary samples taken after at least a 12-hour fast?
  4. As stated previously, “Blood specimens were analyzed for biochemical data using an automatic chemistry analyzer (Beckman DXC880i, Brea, CA, USA) following standardized laboratory procedures.” Is the subsequent detection carried out using a blood sample or a serum sample?
  5. What type of portable sensing system is used to determine the electrical conductivity of salivary specimens collected?
  6. As written, “…the conductivity test could be achieved 140 with a saliva specimen at an extremely low volume (50 uL).”  How was the volume of the smallest 50 uL determined?
  7. How is the selectivity of a sensor determined?
  8. How long is the saliva solution stable during detection?
  9. How long does it take for detection to occur? How it was established? The sensing principle of the used sensor is unknown.
  10. Authors should also discuss the proposed sensor's reusability.

Reviewer 2 Report

In this study, the authors measured salivary conductivity and investigated the relevance in patients with chronic kidney disease. Based on the noninvasive measurement, this study is potentially worth publishing, however several improvements are required.

  • Please add the criteria for hypertension, diabetes, dyslipidemia and gout.
  • I suggest the authors to verify the adequacy of the sample size by means of a power calculation.
  • Because salivary electrolytes may affect the conductivity, serum electrolyte status should be reported.
  • Urinalysis is not performed in this study, therefore the other criteria for diagnosing CKD is missing.
  • There are only 214 subjects in this study. The authors applied too much factors as explanatory variables in the multivariate analysis.
  • Did the authors considered multicollinearity in performing multivariate analyses?
  • Which ROC curve is relevant for the cut off value of high and low salivary conductivity (6.59 ms/cm)?
  • Figure 2B; Why the authors divided the patients at 75 years of age?
  • Subjects with High salivary conductivity had higher proportion of diabetes and hypertension, that are strongly related to the development of CKD. I wonder if salivary conductivity reflected the comorbid conditions but not kidney dysfunction.

Round 2

Reviewer 1 Report

Some comments are not justified. Like,

1. Comment 7, How is the selectivity of a sensor determined? It is very important for any kind of sensor.

2. Authors should also modify the figure 1 and present in a better way. 

3. Also, put the original pictures of developed sensor.

Overall, results and originality is not convincing to be publish in Biosensors Journal. Authors should modify it. 

Reviewer 2 Report

I have no specific comment.

Author Response

Thank you very much for your comments. We are pleased the reviewer is satisfied with our responses.